# Strategies for Transboundary Swine Disease Management in Asian Islands: Foot and Mouth Disease, Classical Swine Fever, and African Swine Fever in Taiwan, Japan, and the Philippines

**DOI:** 10.3390/vetsci11030130

**Published:** 2024-03-15

**Authors:** Chia-Hui Hsu, Chia-Yi Chang, Satoshi Otake, Thomas W. Molitor, Andres Perez

**Affiliations:** 1Center for Animal Health and Food Safety, College of Veterinary Medicine, University of Minnesota, Saint Paul, MN 55108, USA; molit001@umn.edu (T.W.M.); aperez@umn.edu (A.P.); 2School of Veterinary Medicine, National Taiwan University, Taipei 10617, Taiwan; chiayichang@ntu.edu.tw; 3Swine Extension & Consulting, Inc., Niigata 957-0021, Japan; satoshiotake@swext-consulting.co.jp

**Keywords:** transboundary disease, foot and mouth disease, classical swine fever, African swine fever, disease control

## Abstract

**Simple Summary:**

This comprehensive review explores how Taiwan, Japan, and the Philippines manage the threat of swine transboundary animal diseases (TADs), focusing on foot and mouth disease (FMD), classical swine fever (CSF), and African swine fever (ASF). The challenge at hand is the effective control and prevention of these diseases, crucial for safeguarding the swine industry and food security in Asian islands. This study examines varied government strategies, such as mass vaccination, zoning, and intensive surveillance, employed by these islands. Taiwan successfully eradicated FMD through widespread vaccination, Japan effectively controlled CSF re-emergence in wild boars, and the Philippines utilized a zoning strategy for FMD eradication. Despite these successes, the persistent threat of an ASF pandemic, particularly in the Philippines, underscores the need for tailored and unified approaches in transboundary disease control. The lessons gained from these past experiences contribute to global efforts to manage TADs, emphasizing the importance of multi-sectoral and multi-institutional cooperative strategies for the benefit of society’s health and food security.

**Abstract:**

Swine transboundary diseases pose significant challenges in East and Southeast Asia, affecting Taiwan, Japan, and the Philippines. This review delves into strategies employed by these islands over the past two decades to prevent or manage foot and mouth disease (FMD), classical swine fever (CSF), and African swine fever (ASF) in domestic pigs and wild boars. Despite socio-economic differences, these islands share geographical and climatic commonalities, influencing their thriving swine industries. Focusing on FMD eradication, this study unveils Taiwan’s success through mass vaccination, Japan’s post-eradication surveillance, and the Philippines’ zoning strategy. Insights into CSF in Japan emphasize the importance of wild boar control, whereas the ASF section highlights the multifaceted approach implemented through the Philippine National ASF Prevention and Control Program. This review underscores lessons learned from gained experiences, contributing to a comprehensive understanding of swine disease management in the region.

## 1. Introduction

Foot and mouth disease (FMD), classical swine fever (CSF), and African swine fever (ASF) are relevant transboundary animal diseases (TADs) of swine reportable to the World Organisation for Animal Health (WOAH, formerly known as OIE). These highly contagious TADs can rapidly spread across national borders, imposing severe financial and economic impacts to affected regions and countries [1]. 

Particularly concerning is ASF, presenting a formidable challenge in East and Southeast Asia, with elusive characteristics and significant repercussions for the pig industry. The ASF Asian pandemic, affecting China, Mongolia, Vietnam, Cambodia, North Korea, Laos, Myanmar, the Philippines, South Korea, Timor-Leste, Indonesia, Papua New Guinea, India, Malaysia, Bhutan, Thailand, Nepal, and Singapore since August 2018, has caused substantial socio-economic losses and public health impacts in the region. The presence of both genotype I and II ASFVs [2], alongside predominant genotype II outbreaks, poses a global challenge to the swine industry. Recognized by the WOAH as a critical concern, ASF has become the primary focus for swine health surveillance, prevention, and control efforts in the region [3].

The overarching objective of this review is to support ASF prevention and control in the region by leveraging the lessons learned from past experiences on the control of significant TAD incursions over the past two decades. By analyzing demographic and agricultural statistical data, this review acknowledges the socio-economic, linguistic, and cultural variations among Asian islands. It then explores the specific details of the FMD and CSF cases, highlighting successful eradication, long-term control, re-outbreak management, and surveillance efforts. By exploring diverse government strategies for FMD, CSF, and ASF, this review provides insights to inform future efforts in bolstering ASF control. The ultimate goal of this paper is to support the collective commitment of the region, reflecting a shared goal intended to develop effective policies and measures to mitigate ASF’s impact in island territories of Asia.

## 2. Commonalities and Disparities

### 2.1. Commonalities in Geography, Climate, and Swine Industries

Japan, the Philippines, and Taiwan are island territories that share a common archipelagic geographical trait. Japan has around 7000 islands, while the Philippines has about 7600 islands, and Taiwan has 21 islands, including the Penghu archipelago, Kinmen, and Matsu. This shared island identity not only shapes their geographical landscapes but also carries implications for their respective swine industries. One noteworthy aspect is the steady growth and intensification of the thriving swine industries in these islands, contributing to their comparability. Before the ASF outbreak in 2019, the Philippines experienced consistent growth in their swine inventory (Appendix A). Simultaneously, Japan and Taiwan have demonstrated sustained increases in the proportion of larger-scale farmers, with swine holdings exceeding 200 heads over the past two decades.

Additionally, they all share a common subtropical or tropical climate, a significant factor influencing their swine industries. Environmental sustainability plays a crucial role in swine facility management [4]. They require specific measures to prevent overheating and ensure the well-being of the swine population, such as effective ventilation and cooling systems. 

It is noteworthy that Taiwan, despite having a smaller total swine inventory since the 2000s, secured the position of the world’s second-largest pork exporter in 1996. It boasted a sow population of about 1.4 million and approximately 10.7 million on-farm pigs [5]. This information suggests that despite differences in scale and inventory, these three territories have unique dynamics and achievements in the swine industry that can be analyzed together for a comprehensive understanding.

### 2.2. Commonalities in Data Transparency

In comparing the three islands—Taiwan, Japan, and the Philippines—data transparency plays a crucial role as a foundational element. Each of them provides annual or seasonal updates on agricultural information through their respective government ministries. Taiwan, for instance, publishes its annual agricultural report on the Ministry of Agriculture website (https://eng.moa.gov.tw/, accessed on 14 Jan 2024). Japan updates information on agriculture, forestry, and fisheries through its Ministry of Agriculture, Forestry, and Fisheries, as accessible on their official website (https://www.maff.go.jp/e/, accessed on 14 Jan 2024). Similarly, the Philippines Swine Situation Report updates swine production and statistics data, along with other livestock statistics, accessible on the Philippines Statistics Authority (PSA) website (https://psa.gov.ph/, accessed on 14 Jan 2024). Similarly, the Philippines Swine Situation Report delivers swine production and statistics data, along with other livestock statistics, accessible on the PSA website. The inclusion of these reliable sources adds credibility and depth to the research analysis.

This readily available documentation facilitates a comprehensive comparative analysis in this review. Additionally, the recorded Memorandum Circular related to ASF policy in the Philippines enhances transparency in disease situation tracking and recent policy changes. This common feature underscores the shared trajectory of their swine industries and adds to the comparability of their respective policy backgrounds, especially in the formulation of measures addressing swine transboundary diseases.

### 2.3. Disparities in Socio-Economic Status and Transboundary Animal Disease Dynamics

One of the important distinctions among Japan, the Philippines, and Taiwan lies in the realm of socio-economic differences (as indicated by their Gross Domestic Product or GDP and levels of industrialization) significantly influencing the maturity and management of their corresponding swine industries and veterinary resources. These economic variations play an important role in determining the scale and scope of the swine industries, thereby shaping the overall landscape of pig farming. For instance, Japan boasts the third-largest economy globally, with a substantial nominal GDP of USD 5.15 trillion—approximately 8.7 times that of Taiwan and around 14 times that of the Philippines (Appendix B). This economic contrast reveals disparities influencing swine industry development and resources across these nations.

Another important differentiating factor lies in the presence or absence of TADs. Given the threat of the ASF pandemic in Asia since its first appearance in 2018 [6], it is essential to comprehend the prevalence and management of FMD and CSF to evaluate the resilience and vulnerability of swine industries in each island (Figure 1). Disparities in disease control measures (Table 1) and response strategies give rise to distinct challenges and opportunities within their pig farming sectors. In addition, cultural differences further contribute to divergent policy compliance in the swine industry. Unique cultural practices and traditions also influence how policies related to swine farming are formulated and implemented. These differences in policy compliance not only reflect the cultural nuances but also impact the overall effectiveness and adaptability of regulatory frameworks within the swine sector.

## 3. Strategies

### 3.1. FMD Eradication

In Southeast and East Asia, FMD has historically manifested with diverse serotypes, including serotypes A, Asia 1, O, and C [7]. Between 1976 and 1994, the Philippines experienced outbreaks primarily associated with serotype C [7]. However, in the last two decades, significant focus has been directed towards the eradication program, specifically targeting the PanAsia strain of serotype O in Japan, the Philippines, and Taiwan [8]. This emphasis arises from recognizing that serotype O is adapted to porcine hosts in these regions, leveraging the advantages of island geography and the prevalence of a single genotype.

Currently, while FMD remains endemic in many parts of Southeast and East Asia, Japan, the Philippines, and Taiwan have successfully maintained FMD-free status with non-vaccination practices. Japan and the Philippines achieved this status in 2011, whereas Taiwan attained it in 2020 (Figure 1).

#### 3.1.1. FMD in Taiwan

In 1997, Taiwan faced a serious FMD outbreak that likely originated from the smuggling of pigs. The outbreak escalated quickly, spreading to 20 counties and cities in Taiwan, prompting swift government action. To combat the FMD outbreak, the government established the “Foot-and-Mouth Disease Eradication Task Force” in March 1997. Their approach involved widespread preventive vaccinations and the stamping out of FMD-infected pigs [8,9,10]. Although this strategy brought the situation under control in four months, it came at a cost—over 4 million pigs were slaughtered, and economic losses exceeded NTD 170 billion (approximately USD 5.7 billion) [11].

Recognizing the need to assess the infection status of swine populations after mass vaccination, a serological surveillance program was established in April 2000. This program monitored the immune and disease status of pig populations using a standard virus neutralization test and relied on the detection of antibodies to FMD virus Non-Structural Proteins (NSPs) 3ABC, 3AB, and 3B for differentiating between infected and vaccinated animals [8].

By May 1997, the epidemic was effectively contained, stabilizing domestic pork prices. Taiwan remained FMD-free for over two years after the last reported case in February 2001, earning recognition from the WOAH as “FMD-free with vaccination.” Despite this accomplishment, ongoing risk assessments revealed the virus’s persistence in the field, posing a continuing threat [12]. Secondly, the allure of the global market served as a strong incentive, prompting the government’s commitment to pursue the status of the “FMD-free without vaccination” status from the WOAH to rejoin the global pork market.

While attempts to cease vaccination in 2003 and 2009 were unsuccessful, East Asia experienced another massive outbreak of FMD in 2010 [13]. During this period, Taiwan maintained a risk-based FMD vaccination strategy, persisting in its policy of fully administering FMD vaccines. Taiwan saw no FMD cases after 2013, leading to continuous vaccine use. The third attempt to halt vaccination began on 1 July 2018, resulting in an absence of FMD cases by 30 June 2019. In September 2019, Taiwan, Penghu, and Matsu applied to the WOAH for reclassification as an FMD-free zone without vaccination, and they received approval in June 2020 [14].

This success was driven by three key factors: (1)Consistently achieving high swine vaccination coverage, exceeding the targeted annual rate of 90%. In 2015 and 2016, the pig vaccination rate reached 111.09%, the herbivorous animal vaccination rate was 106.39%, and the overall vaccination rate stood at 110.89%, according to government data [15].(2)Utilizing NSP ELISA tests for sero-surveillance post-2009 FMD recurrence struck a balance between sensitivity and specificity. This ongoing testing aims to substantiate the absence of the FMD virus and guide decisions on lifting compulsory vaccination mandates.(3)Dedicated personnel overseeing immunization certificates in meat markets, along with routine veterinary inspections, reinforced swine immunization certificate collection, environmental disinfection, pig movement management, and timely epidemic reporting.

In summary, the 1997 FMD virus outbreak severely impacted Taiwan’s swine industry, influencing industry transformation and government policy awareness regarding transboundary animal diseases. Despite a pre-outbreak swine population of 10.7 million, ongoing repercussions resulted in a 50% decline with no full recovery. Over 23 years of control efforts, Taiwan achieved success in becoming a WOAH-recognized FMD-free zone without vaccination.

#### 3.1.2. Eradication of FMD in Japan and Ongoing Prevention Initiatives

Foot and mouth disease (FMD), historically absent in Japan since 1908, reemerged in Miyazaki Prefecture and Hokkaido Prefecture from March to May 2000, impacting four cattle farms and causing a significant outbreak. Japan swiftly implemented a stamping-out policy and intensive surveillance, adhering to a non-vaccination strategy. By the end of September 2000, the country successfully regained FMD-free status, marking a pivotal achievement [16].

In 2010, another notable FMD outbreak occurred in Miyazaki Prefecture, the first in approximately a decade [17]. Rapid biosecurity prevention measures, including culling, the burial of infected animals, farm disinfection, and movement restrictions, were enacted. The epidemic, which involved the culling of approximately 290,000 animals, was successfully confined to a localized area in Miyazaki Prefecture and eradicated within a span of three months [17]. By July of the same year, all movement restrictions were lifted. Subsequently, in October 2010, Japan applied for international recognition from the WOAH, resulting in official certification as a “non-vaccination FMD-free country” on February 2011. Maintaining this disease-free status since then, Japan’s ongoing commitment is evident in the absence of reported FMD outbreaks, highlighting successful disease prevention and control.

Despite Japan’s success in eradicating FMD, the ongoing threat persists, particularly from neighboring Asian countries like South Korea, North Korea, Russia, and China. The Ministry of Agriculture, Forestry, and Fisheries (MAFF) remains actively engaged in comprehensive measures, including stringent inspections and disinfection protocols at international airports and ports for importing animal and livestock products. Emphasizing hygiene management and early detection, MAFF encourages collective vigilance and urges the public to adopt maximum precautions, enhancing FMD surveillance measures [18]. The government remains steadfast in fortifying domestic epidemic prevention measures, ensuring a proactive and effective defense against potential FMD spread within Japan.

#### 3.1.3. Philippines: FMD Eradication through Zoning Strategy

The Philippines eradicated FMD through a zoning strategy with three distinct phases: control (1996–2000), consolidation (2000–2004), and eradication (2004–2009) [7,19]. The approach involved effective animal movement management, intensive monitoring through active and passive surveillance, and strategic mass vaccination in high-risk areas. For the first stage, the government adopted a progressive zoning approach, classifying regions based on their FMD status [20]. These zones, distinguished by colors ranging from red to green, specified restrictions on the movement of live pigs and pork-related products. Therefore, intensive monitoring stands out as a crucial factor in FMD control in the Philippines. Active surveillance involved serological surveys and regular clinical monitoring on farms, stockyards, and inspections conducted at various veterinary quarantine facilities. Passive surveillance encompassed the submission of monthly negative monitoring reports.

Strategic mass FMD vaccination in high-risk areas, including swine farms, livestock markets, holding yards, and abattoirs, played an important role in tackling the endemic. However, challenges included low herd immunity, farmers’ reluctance to vaccinate pregnant pigs, and issues with vaccination techniques and efficacy [20]. Relying solely on vaccination is not sufficient. Integrating mass vaccination with efforts to enhance public awareness and train livestock industry personnel emerges as crucial in successfully eliminating FMD. With no reported FMD outbreaks for over three years, the formal withdrawal of FMD vaccination was implemented through the AO No. 12 series of 2009, marking a significant milestone in eradicating the transboundary threat in the Philippines.

#### 3.1.4. FMD as a Regional Threat

Because, as of 2023, FMD is still prevalent in other countries in the region, island territories face a persistent FMD threat from neighboring countries. Recent outbreaks in mainland Asia, including South Korea (11 cases of FMD serotype O recurrence in May 2023) and China (two new FMD outbreaks in Guangxi and Xinjiang, affecting over 200 animals, caused by FMD serotype O) serve as an example of risk for transboundary disease spread into the islands [21,22]. To preserve FMD-free status, islands must prioritize stringent border controls at airports and ports, emphasizing inspections and disinfection. Encouraging livestock industry personnel to uphold strict feeding hygiene and detect potential outbreaks early is also crucial.

### 3.2. Re-Emergence of CSF in Japan

#### CSF in Wild Boar Re-Emergence in Japan

Despite achieving CSF eradication and obtaining official recognition as CSF-free (Figure 1) in 2015 by the WOAH [23], Japan experienced a re-emergence in 2018. The resurgence affected farrow-to-finish farms in Gifu Prefecture, confirmed by laboratory tests at the National Institute of Animal Health, National Agriculture and Food Research Organization (NIAH-NARO), and ended a 26-year CSF-free period [24]. The isolated pathogen differs from the strain that caused the preceding CSF outbreak before the 2015 eradication. Identified as genotype 2.1, this causative pathogen is known for triggering recurrent outbreaks in East and Southeast Asia [25]. The re-emergence in 2018 is under investigation in connection with tourists who brought in meat-related products contaminated with CSF from China or other Southeast Asian countries and subsequently tested positive for CSF.

Despite the setback, Japan’s swine population remains substantial at 9.29 million heads as of 2021 (Appendix A), indicating minimal impact on the pork industry and swine production in recent years [26]. The outbreak initially saw positive rates of total animals tested for CSF, ranging from 10% to 20%, spiking between March and June 2019 before stabilizing around 10% in the second half of 2019 [27]. All 1110 notifications of CSF outbreak cases from 2018 to August 2019 were localized to Gifu Prefecture and its surrounding four prefectures (Figure 2), involving 1071 cases in wild boars and 39 in domestic pig farms [28]. Spatiotemporal models suggested that CSF likely spread through the movement of wild boars, highlighting the importance of monitoring and assessing the risk to swine farms [29].

In response to the CSF epidemic, the Japanese government, in collaboration with various ministries and stakeholders, implemented robust countermeasures to address disease spread in both wild boars and domestic pigs. MAFF took a comprehensive approach, intensifying surveillance efforts in regions where CSF-affected wild boars were identified. In March 2019, an oral vaccine, commonly referred to as a bait vaccine, was strategically administered to wild boars in selected prefectures with positive CSF cases and in surrounding areas [30]. A study on CSF dynamics and wild boar bait vaccination in Japan found a 12.1 percentage point increase (95% CI: 7.8–16.5) in antibody prevalence from the 2019 vaccine campaigns. The findings underscore the potential need for larger bait vaccine distribution campaigns, considering uncertainties in the basic reproduction number (R_0_ = 1.5–2.5) and target vaccination coverage [31]. Distributing CSF vaccines widely enough in wild boars’ habitats is crucial for enhancing oral vaccination effectiveness. A study highlights that pre-launch investigations within a 1200 m radius of the distribution point are essential to assess the wild boar population [32].

The establishment of vaccination belts in both eastern and western Japan was a key component of these efforts [33]. Collaborating closely with relevant agencies such as the Ministry of Environment, the National Police Agency, the Fire and Disaster Management Agency, and the Ministry of Defense, the MAFF aimed to proactively manage the situation [33]. The comprehensive strategy included intensive hunting, capturing, surveillance, fencing, disinfecting areas with wild boar contact, and the application of oral vaccination through bait [34]. Together, these reflect a concerted effort to control and mitigate the impact of CSF in wild boar populations.

Despite sporadic CSF cases persisting in wild boar in 2023, Japan maintains transparency through detailed documentation on the MAFF website, including case index numbers for effective disease management. Vaccination has reduced CSF transmission from wild boars to domestic pig farms [35]. After losing WOAH CSF-free certification in 2020, Japan implemented preventive measures, focusing on virus restriction, wildlife management, hygiene zones, cleaning protocols, and dedicated clothing. Combining efforts to prevent wildlife intrusion with strategic vaccination and preventive measures forms a comprehensive approach against CSF. Japan’s effective containment of CSF is attributed to robust surveillance, ongoing research, and evidence-based biosecurity policies.

### 3.3. Five Years of ASF Experience in the Philippines and Ongoing Threats

Within the scope of TADs examined in this review, ASF emerges as the most formidable threat to Asian countries due to its high mortality rate and the absence of an effective vaccine. The swine industry faced an unprecedented threat with the onset of the 2018 ASF outbreak in China and neighboring countries. These outbreaks led to considerable economic losses, affecting more than 18 Asian countries [6]. Additionally, ASF outbreaks were reported in both wild boars and domestic pigs.

The ASF impact on the Philippines is illustrated in Figure 3 and Table 2, depicting the spread of ASF from its initial detection on Luzon Island to all 17 regions by the end of 2023. This resulted in significant economic setbacks and increased the pork retail price. The Philippines government has implemented a comprehensive approach to combat ASF, including the 1-7-10 protocol (1 km: cull pigs near infected farms; 7 km: monitor and test actively; 10 km: swine farms report surveillance), movement control measures, and the initiation of the National African Swine Fever Prevention and Control Program (formerly known as Bantay ASF sa Barangay). Initiated by the Philippine College of Swine Practitioners, this program has three primary goals. Firstly, it aims to ensure business continuity for both smallholder and commercial swine farmers. Secondly, it emphasizes the use of point-of-care tests, including proactive surveillance using PCR and the ASF Rapid Test kit, to effectively mitigate the impact. Thirdly, the program focuses on facilitating repopulation efforts. Through the collaboration of various stakeholders, the program continues to perform.

While the Philippines has attempted to implement a progressive control strategy similar to that employed and improved upon in historical FMD control, the progress has been limited. The challenges encountered in controlling ASF in the country include a shortage of trained manpower, the intricacies of the swine value chain (lack of traceability), and the persistence of high-risk practices in specific farming sectors and live animal trade [36].

In Taiwan and Japan, the primary risk of ASF introduction is associated with border control inspections, as studies indicate ASFV can be detected in pork products brought by travelers [37,38,39,40]. Empirical evidence from Taiwan reveals a notable increase in the detection of illegal pork products carried by international travelers, particularly peaking in January and February [39]. Despite pigs in domestic farms testing negative for ASF in both Taiwan and Japan, a persistent threat arises from travelers transporting pork-related products that may harbor ASFV. As of December 2023, Taiwan’s Veterinary Research Institute reported 79.3% of 1733 cases of illegal meat transportation from China [41], with a high 12.4% positive rate for ASFV, indicating a worsening situation with no improvement signs. Regarding punishment, passengers carrying pork products into Taiwan from ASF-affected areas within the past three years will face fines: first offense: NTD 200,000 (USD 7000); subsequent offenses: NTD 1,000,000 (USD 35,000). Foreign nationals failing to pay on the spot will be denied entry and deported [42]. Heightened surveillance efforts, particularly in airports and seaports [40], underscore the importance of citizens’ compliance as a key factor in preventing infection routes in Taiwan. This highlights the critical need for stringent monitoring and preventive measures to eliminate ASF spread.

## 4. Discussion and Lessons Learned

### 4.1. FMD

In the pursuit of long-term FMD eradication, diverse strategies were employed. Taiwan places a strong emphasis on achieving high vaccination coverage and intensive surveillance. In contrast, the Philippines serves as an exemplary model by effectively leveraging various components to collaboratively work towards eradication goals, even in the absence of high herd immunity. Despite potential imperfections in the FMD vaccine and challenges in attaining the desired herd immunity, the Philippines successfully combines essential pillars. These include evaluating serological responses in vaccines, zoning controls, enhancing government surveillance and emergency response capabilities, and raising public awareness of biosecurity measures. The integration of these policies significantly contributes to halting outbreaks and ultimately eradicating FMD. Japan’s swift response to the FMD reappearance in 2010 featured measures like culling, burial, and farm disinfection and localized the epidemic, enabling progressive containment and eventual eradication without vaccines. This post-eradication success serves as a lesson for island territories, highlighting the effectiveness of timely and intensive control through rapid biosecurity measures. The varied approaches of different countries underscore the achievement of FMD eradication goals with unique resources and capabilities.

### 4.2. CSF in Wild Boar Control

The re-emergence experience of CSF in Japan emphasizes the critical importance of controlling transboundary diseases in wild boars and raises concerns about potential implications. (1) Despite achieving eradication, the persistent risk of new introductions or new strains of CSF from other countries remains. (2) Addressing CSF in wild boars poses an additional challenge to consider.

In the case of ASF, it could also manifest in wild boars, particularly in the three countries with native wild boar populations in their forests [43]. The increasing interaction between humans and wildlife elevates the risk of pathogen transmission from wild boars to domestic pigs, especially in backyard systems, where direct contact between domestic pigs and wild boars complicates disease control efforts. Despite sporadic cases of CSF persisting in Japan after extensive vaccination bait and oral vaccination campaigns by the government, the experience has highlighted the resilience of the swine industry. Business continuity remains robust, and the overall stability of the swine inventory not only reflects this resilience but also suggests a positive outlook for ongoing swine disease control efforts in the country.

### 4.3. Smallholder Pig Raising Still Poses a Potential Risk

In backyard or smallholder settings, certain characteristics are notable in these countries. Firstly, there is a tendency to purchase live pigs of poor quality [36]. Additionally, the feeding of pigs with swill is a common practice, despite being prohibited or not recommended by the government. Moreover, there is a lack of strict adherence to biosecurity measures, and a shortage of resources for veterinary service notification is observed. Although smallholder settings have decreased compared to the past decades (Taiwan at around 33% and Philippines at 67.5% as per Appendix B), increasing public awareness and empowering veterinary services globally remains crucial.

### 4.4. Potential CSF Risk and ASF Prevention 

Since the 2019 ASF outbreak, the Philippines has primarily addressed ASF, neglecting enzootic CSF. Limited research on CSF further complicates the understanding of its prevalence and transmission in the islands. It is worth noting that while potential economic losses of ASF are estimated [44], estimates for CSF in the Philippines are not available. The confusion between these two viral diseases, coupled with the misuse of vaccinations, underscores the need for clear regulations on CSF control and differential diagnosis. The current status of CSF in the Philippines is uncertain, demanding urgent attention.

The spreading pattern and control policies employed for ASF in the Philippines serve as instructive lessons for Japan, Taiwan, and regions currently free from ASF. In the Philippines, the movement and zoning policies constitute a central systemic approach to outbreak management, though achieving elimination remains a substantial challenge, as seen in infections across all regions over the past five years. Additionally, a recurrent pattern of ASF outbreaks is observed on both spatial and temporal scales [45]. While it is uncertain whether the same dynamics apply to Japan and Taiwan, further research on ASF in the Philippines can significantly contribute to enhancing disease control knowledge in these island countries.

In December 2023, Taiwan’s Veterinary Research Institute detected a new virus in pork products at airport inspection, formed through a genetic recombination of ASFV genotypes I and II. This alarm underscores the crucial role of border inspection in addressing transboundary animal disease threats, which is key for the protection of island territories.

### 4.5. ASF Vaccine Practicality: Lessons from FMD and CSF, Managing Expectations, and Prioritizing Prevention

Lessons from the eradication of FMD and CSF have highlighted the benefits of vaccination in managing these TADs. Some of the ASF policies also share common characteristics with FMD in the Philippines, especially regarding zoning and movement policies. This previous success has sparked optimism among the people of the Philippines, encouraging vaccine trials in the field to harness the advantages of commercially available ASF vaccines. However, it is important to note that the ASF vaccine, though commercialized with products [46] like NAVET-ASFVAC and AVAC ASF LIVE, is not a flawless solution and has its limitations. Reliance on live attenuated strains may not provide sterilizing immunity [47], allowing vaccinated pigs to still become infected if exposed to wild ASF strains or other genotypes. Safety, although demonstrated under experimental conditions, poses potential adverse effects in the field. Current evaluations are limited to weaned piglets with specific age recommendations for inoculation. Furthermore, lower-virulence ASFV genotypes I and II were recently identified in China in June 2021 [48]. The ongoing ASF outbreak in China involves both genotypes, which are not addressed by the commercial live attenuated vaccine derived from genotype II ASFV [2]. If chronic ASFV infection spreads, disease control becomes more challenging [49].

### 4.6. A Vaccination Trade-Off

The utilization of vaccines poses a complex dilemma for governments, necessitating consideration not only of public health concerns but also the intricacies of the international pork trade and compliance with global organizations. Take Taiwan’s FMD eradication efforts as an example. The use of vaccines has proven advantageous in enhancing the FMD-negative status [12], with the associated costs remaining manageable for the government, ensuring affordability. Furthermore, farmers exhibit compliance through achieving high vaccination rates.

However, the government must weigh the long-term implications on the international pork trade to fully participate in the global pork market. Through thorough discussions, acknowledging the significance of global trade, the Taiwanese government opts to persist in the long-term eradication of FMD to safeguard the pork industry. This dilemma applies to three TADs, crucial for global pork market participation. Similarly, the use of ASF vaccines demands careful consideration, striking a balance between disease control and economic factors to navigate potential international trade restrictions.

### 4.7. GF-TADs

The imperative to establish a comprehensive, multi-sectoral, and multi-disciplinary approach in addressing TADs has gained prominence, particularly in the aftermath of the global impact of the COVID-19 crisis. The Global Framework for the Progressive Control of Transboundary Animal Diseases (GF-TADs) emphasizes enhanced control for global health, involving the establishment of strategies for priority TADs at regional and sub-regional levels [3,50]. This includes strengthening engagement and coordination with various countries, stakeholders, private sectors, and enhancing the sustainability of TAD partnerships. The regional insights from Taiwan, Japan, and the Philippines in this review offer valuable learning experiences, setting the stage for effective strategies and collaborations on a broader scale.

### 4.8. Unified Measures for Transboundary Animal Disease Control

Adopting a comprehensive perspective that considers FMD, CSF, and ASF collectively has been implemented in Taiwan’s national border controls. For example, enhancing airport inspections not only mitigates the risk of ASF introduction but also reduces transmission risks of CSF and FMD through international travelers. Plus, the prohibition and regulation of swill feeding on farms could eliminate the potential risk for all swine transboundary diseases. This comprehensive approach, encompassing early detection, disease management, and risk analysis, significantly alleviates the risks associated with these TADs. Effective communication between governments and compliance by farmers and stakeholders will also be crucial. Taiwan has ceased nationwide CSF vaccinations as of July 2023 and is actively engaged in collective surveillance for FMD, CSF, and ASF. An application for Taiwan to be recognized as a CSF-free region will be submitted to the WOAH after June 2024 [51].

## 5. Conclusions

The lessons from each island provide comparative advantages. Examining transboundary swine disease management in Taiwan, Japan, and the Philippines reveals shared challenges and diverse strategies, especially amid the ASF pandemic in Asia. Despite socio-economic differences, commonalities like tropical climate and transparent documentation offer a comparative advantage. Taiwan’s success in FMD eradication through mass vaccination, Japan’s post-eradication surveillance for FMD, and the Philippines’ zoning strategy provide valuable lessons. Experiences with CSF and ongoing ASF prevention efforts highlight the importance of tailored approaches. Similar policies for transboundary disease prevention suggest a unified approach. Addressing CSF and ASF in the Philippines, maintaining CSF control in wild boar in Japan, and Taiwan’s pursuit of CSF-free status shape future directions.

## Figures and Tables

**Figure 1 vetsci-11-00130-f001:**
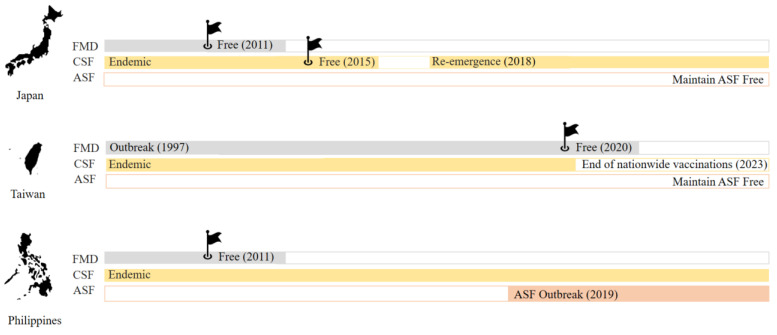
An overview of the current status of African swine fever (ASF), foot and mouth disease (FMD), and classical swine fever (CSF) in Japan, Taiwan, and the Philippines.

**Figure 2 vetsci-11-00130-f002:**
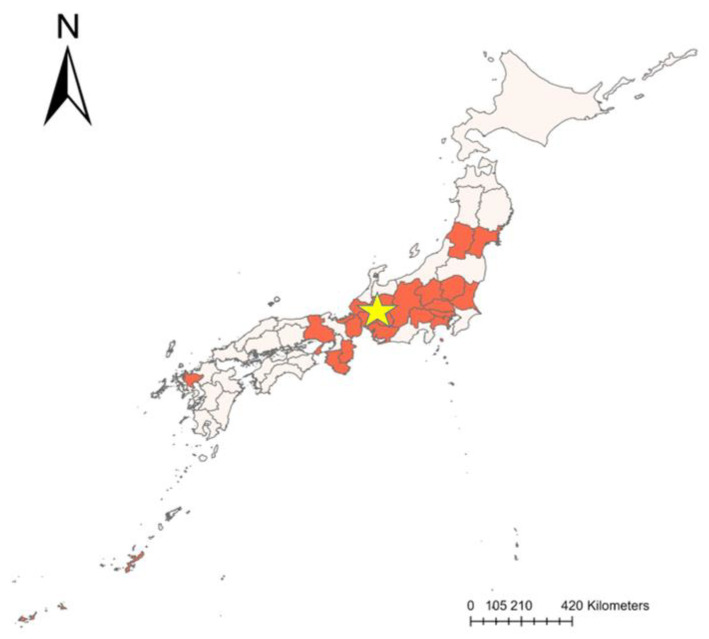
CSF re-outbreaks in Japanese prefectures (2018–2023) starting from the Gifu Prefecture outbreak in September 2018 and continuing until December 2023; a total of 89 events were reported across 20 prefectures, resulting in the culling of approximately 368,000 animals. The red regions on the map indicate the infected prefectures, with the yellow star specifically highlighting Gifu Prefecture.

**Figure 3 vetsci-11-00130-f003:**
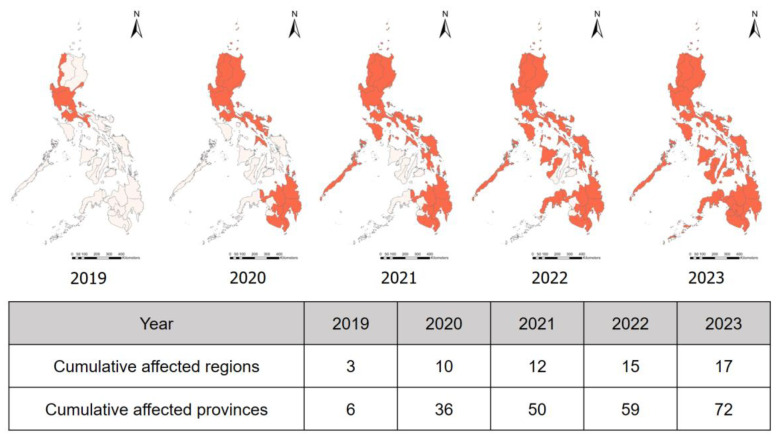
The cumulative progression of African swine fever (ASF) in the Philippines from 2019 to 2023. The map showcases the spread of ASF, originating from Luzon Island and progressively expanding to encompass all 17 regions by December 2023. The red regions on the map represent the affected administrative level.

**Table 1 vetsci-11-00130-t001:** A policy overview summarizing the focused strategies adopted by Japan, the Philippines, and Taiwan to address foot and mouth disease (FMD), classical swine fever (CSF), and African swine fever (ASF) in the context of this review.

Territories	FMD (Historically Focused Strategy)	CSF (Current Focus)	ASF (Current Focus)
Japan	Stamping-out policy and intensive surveillance, adhering to a non-vaccination strategy.	Larger-scale bait vaccination in wild boar.	Strengthen inspections at border control.
Philippines	Multiple collaborative strategies, including zoning and strategic vaccination in high-risk areas.	Conduct routine surveillance and differentiate between ASF and CSF.	Implement National African Swine Fever Prevention and Control Program.
Taiwan	High swine vaccination coverage and regular sero-surveillance.	Cease CSF vaccination with sentinel animal monitoring.	Strengthen inspections at border control.

**Table 2 vetsci-11-00130-t002:** Timeline of ASF in the Philippines: key events and developments. This table is referenced from the Department of Agriculture (DA) governmental information.

Date	Event
July 2019	The first ASF outbreak in a swine backyard farm was reported in the Philippines.
September 2019	The Department of Agriculture (DA) confirmed ASF presence in Rizal and Bulacan provinces on Luzon island.
December 2019	The DA issued Administrative Circular 12 (National Zoning and Movement Plan for the Prevention and Control of African Swine Fever), formally ordering the establishment of zones across the country depending on the level of risks of regions in relation to ASF.
January 2020	The first recorded ASF outbreak in Mindanao island.
August 2020	ASF cases peaked with 1773 positive reports.
January 2021	First ASF case confirmed in Leyte province, Eastern Visayas.
Feb 2021	The Bantay ASF sa Barangay Program (BABay ASF) was launched.
May 2021	President Duterte declared a state of calamity in relation to the ASF epidemic.
March 2023	ASF outbreak in Cebu (Central Visayas, Region VII), after three years of being an ASF-free region, posed a threat to one of the largest hog industries in the Visayas.
July 2023	Initial local trials for an ASF vaccine in the Philippines.

## Data Availability

Data contained within this article.

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
