# Peer review of "Strategies for Transboundary Swine Disease Management in Asian Islands: Foot and Mouth Disease, Classical Swine Fever, and African Swine Fever in Taiwan, Japan, and the Philippines"

_vetsci, 2024, doi:10.3390/vetsci11030130_

Round 1

Reviewer 1 Report

Comments and Suggestions for Authors

The title of the paper should be reformulated

Strategies for Transboundary Swine Disease Management in Asian islands: Footand Mouth Disease, Classical Swine Fever and African Swine Fever (Taiwan,Japan, Philippines)

Line 64-68 the paragraph need to shortened, not necessary so many geographical details authors should focus on releant topic: swine production characteristics in selected countries.

Line 121-125 not neccesary to repeat the data which are easy visible in the Figure 1. It can be preformulated but do not need to repaet it.

Generally, the paper need to be shortened, not neccesary to repaet data and give so many country-specific details. The authors need to focus of relevant factors that influence regional diseases spread and socific country measures.

Comments on the Quality of English Language

The english language quality is good, only minor mistakes are required to edit.

Author Response

  1. For the title of this paper, we genuinely value reviewer#1 suggestions. However, it's important to consider that the other two reviewers haven't suggested this change. Due to our preference for the original title we proposed, we kindly ask to keep it unchanged for this manuscript. Nonetheless, we are open to considering the Editor's perspective if they have a different view.
  2. Lines 64-66: These paragraphs have been condensed into two lines, removing unnecessary details, as suggested by the reviewer.
  3. Line 117-121: Sentences were deleted.
  4. We have made efforts to streamline the paper by minimizing country-specific details and focusing more on relevant factors influencing the spread of diseases. Thank you so much for the feedback and suggestion.

Reviewer 2 Report

Comments and Suggestions for Authors

Line 174: coverage according to government data. It would be appropriate that you offer a reference here for that.

Line 185: …. achieved success ….. (with a space between achieved and success)

Lines 188-189 and line 193: it would be helpful for the reader to indicate here presumed introduction route(s)

Line 241: put Figure 2 heading in bold

Line 246-251: the heading indicates CSF in wild boar re-emergence but the text is on affected farrow -to-finish farms, there seems to be no relationship

Line 253-254: … the text reads like tourists who tested positive for CSF, but that is not the precise meaning, it is about tourists probably bringing in CSF contaminated goods/meat, and that is what you should say.

Line 258-259: outbreaks saw CSF positive rates from 10% to 20%. What are these % based on ?? total number of farms or positive pigs within farms ? should be better indicated.

Line 265: use the word epidemic instead of outbreak

Line 285-286: not only mention that it reduced transmission, how much reduction would be informative.

Line 286: the word Despite is used in an incorrect way. When you loose CSF-certification, it seems that it is absolutely necessary to do everything to gain it back by implementing ……

Line 309: …1-7-10 protocol ….. this protocol just jumps into the text without any explanation, would be appropriate to know what this protocol means.

Line 326-330: please add some text indicating what the punishment in terms of a fee or imprisonment there is if a traveller is caught with smuggling

Line 364-365:  the text refers to all kind of risks like live pig trading of low quality, swill feeding, lack of adherence of biosecurity measures……… but the prevalences mentioned in the text do not relate to measurement of insufficiencies, but to the % of small holders. So be more precise !!

Line 370-371: yes, clinical signs of SCF and ASF are identical, but I don’t understand the text that this would lead to CSF case underreporting. Because the signs are not pathognomonic for either CSF or ASF, but signs will and should lead to reporting and then with laboratory diagnostics you know if it is CSF or ASF, so it is not logical to say that it would lead to CSF case underreporting.

Reviewer 3 Report

Comments and Suggestions for Authors

Overall the manuscript carefully describes the measures taken by the authorities of each country to limit the consequences of three extremely significant swine diseases. A few comments that could improve the manuscript are suggested in the attached pdf file

Author Response

Thank you Reviewer#3, for the precious feedback. We have made several revisions here.

  1. Line 147-149: The exact number of farms in 1997 FMD outbreak was not precisely recorded. However, in Line 148, we provide the quantified figure that over 4 million pigs were slaughtered.
  2. Line 163 was revised as suggested by the reviewer.
  3. Lin 181: this sentence was revised.
  4. Line 190: The number of animals has been provided as suggested by the reviewer.
  5. Line 360-362: We have added the importance of empowering the Veterinary Services globally.